# DFM: Interpolant-free Dual Flow Matching

**Denis Gudovskiy**[1]    **Tomoyuki Okuno**[2]    **Yohei Nakata**[2]

[1]Panasonic AI Lab, USA    [2]Panasonic Holdings Corporation, Japan

denis.gudovskiy@us.panasonic.com    {okuno.tomoyuki, nakata.yohei}@jp.panasonic.com

## Abstract

Continuous normalizing flows (CNFs) can model data distributions with expressive infinite-length architectures. But this modeling involves computationally expensive process of solving an ordinary differential equation (ODE) during maximum likelihood training. Recently proposed flow matching (FM) framework allows to substantially simplify the training phase using a regression objective with the interpolated forward vector field. In this paper, we propose an interpolant-free dual flow matching (DFM) approach without explicit assumptions about the modeled vector field. DFM optimizes the forward and, additionally, a reverse vector field model using a novel objective that facilitates bijectivity of the forward and reverse transformations. Our experiments with the SMAP unsupervised anomaly detection show advantages of DFM when compared to the CNF trained with either maximum likelihood or FM objectives with the state-of-the-art performance metrics.

## 1 Introduction

Discrete-time (DNF) and continuous-time (CNF) normalizing flows have been previously extensively studied and compared in details [Ruthotto and Haber, 2021]. With the pros and cons in each approach, we are motivated to apply CNFs in real-world generative and density estimation applications. Theoretically, an infinite-length architecture with arbitrary parameterization can lead to a significant advantages of CNFs when compared to DNF shortcomings. Meanwhile, CNFs with the ordinary differential equation (ODE) integration step endure higher computational complexity, numerical instabilities and approximation errors [Chen et al., 2018]. In particular, the latter is crucial in density estimation which requires accurate estimates of the Jacobian matrix trace.

Recent flow matching (FM) framework [Lipman et al., 2023] simplifies the training phase in CNFs by introducing a new regression objective. This objective minimizes mean square difference of a parameterized vector field model and an interpolated vector field between two data distributions. While the former is a conventional neural network with time-dependent conditioning, the latter relies on certain assumptions about modeled data distributions. As a result, there is an extensive line of research that proposes various forms of the interpolated vector fields and the corresponding probability paths. We summarize recent works in Table 1 with the formal introduction in Section 2. Diffusion models [Song and Ermon, 2019, Ho et al., 2020] that solve stochastic differential equations (SDEs) can also be generalized using the FM framework [Tong et al., 2024].

In this paper, we analyze current interpolation-based FM approach and its inherent limitations i.e. the Gaussian probability path assumption between two data distributions [Lipman et al., 2023]. To address this limitation, we propose a novel *interpolant-free dual flow matching* (DFM) method. Specifically, we accomplish interpolant-free FM using an additional parameterized reverse vector field model. For simplicity, we model the reverse vector field using exactly the same architecture as for the forward one. Then, we optimize our DFM using an objective that enforces transformation bijectivity of the modeled forward and the reverse vector fields.

Table 1: Generalization of probability paths for diffusion and flow matching methods by the [Tong et al., 2024] framework. Unlike ours, these methods rely on an interpolation of the probability paths.

| Probability Path | $q(\boldsymbol{z})$ | $\mu_t(\boldsymbol{z})$ | $\sigma_t$ |
|---|---|---|---|
| Var. Exploding SDE [Song and Ermon, 2019] | $q(\boldsymbol{x}_1)$ | $\boldsymbol{x}_1$ | $\sigma_{1-t}$ |
| Var. Preserving SDE [Ho et al., 2020] | $q(\boldsymbol{x}_1)$ | $\alpha_{1-t}\boldsymbol{x}_1$ | $\sqrt{1-\alpha_{1-t}^2}$ |
| FM [Lipman et al., 2023] | $q(\boldsymbol{x}_1)$ | $t\boldsymbol{x}_1$ | $t\sigma - t + 1$ |
| Rectified FM [Liu et al., 2023] | $q(\boldsymbol{x}_0)q(\boldsymbol{x}_1)$ | $t\boldsymbol{x}_1 + (1-t)\boldsymbol{x}_0$ | $0$ |
| Var. Preserving FM [Albergo and Vanden-Eijnden, 2023] | $q(\boldsymbol{x}_0)q(\boldsymbol{x}_1)$ | $\cos(\pi t/2)\boldsymbol{x}_0 + \sin(\pi t/2)\boldsymbol{x}_1$ | $0$ |
| I-CFM [Tong et al., 2024] | $q(\boldsymbol{x}_0)q(\boldsymbol{x}_1)$ | $t\boldsymbol{x}_1 + (1-t)\boldsymbol{x}_0$ | $\sigma$ |

## 2 CNF Preliminaries and Prior FM Methods

**Continuous normalizing flows.** We follow Lipman et al. [2023] and Tong et al. [2024] notation. We consider a pair of data distributions $q(\boldsymbol{x}_0)$ and $q(\boldsymbol{x}_1)$ over $\mathbb{R}^D$ with densities $p(\boldsymbol{x}_0)$ and $p(\boldsymbol{x}_1)$, respectively. Often, the $p_0 = p(\boldsymbol{x}_0)$ density represents a known prior distribution while the data density $p_1 = p(\boldsymbol{x}_1)$ is not given with only access to an empirical $\hat{q}(\boldsymbol{x}_1)$ and $p_1$ needs to be estimated.

Then, there are a *probability density path* $p : [0,1] \times \mathbb{R}^D \to \mathbb{R} > 0$, which is a time-dependent probability density function $p_t(\boldsymbol{x})$ with $t = [0,1]$ such that $\int p_t(x)dx = 1$, and a Lipschitz-smooth time-dependent *vector field* $u : [0,1] \times \mathbb{R}^D \to \mathbb{R}^D$. The vector field $u_t$ is used to construct a time-dependent diffeomorphism i.e., the CNF $\phi : [0,1] \times \mathbb{R}^D \to \mathbb{R}^D$ that is defined via the ODE as

$$d\phi_t(\boldsymbol{x})/dt = u_t(\phi_t(\boldsymbol{x})) \text{ and } \phi_0(\boldsymbol{x}) = \boldsymbol{x}_0, \tag{1}$$

where $\phi_t(\boldsymbol{x})$ is the ODE solution with $\phi_0(\boldsymbol{x})$ initial condition that transports $\boldsymbol{x}$ from time 0 to time $t$.

On the other hand, $\phi_t$ induces a push-forward $p_t = [\phi_t]_\#(p_0)$ that transports the density $p_0$ from time 0 to time $t$. The time-dependent density $p_t$ is characterized by the *continuity equation* written by

$$\partial p_t(\boldsymbol{x})/\partial t = -\text{div}(p_t(\boldsymbol{x})u_t(\phi_t(\boldsymbol{x}))) = -\text{div}(f_t(\boldsymbol{x})), \tag{2}$$

where the divergence operator, $\text{div}$, is defined as the sum of derivatives of $f_t(\boldsymbol{x}) \in \mathbb{R}^D$ w.r.t. all elements $x_d$ or, simply, the Jacobian matrix trace: $\text{div}(f(\boldsymbol{x})) = \sum_{d=1}^{D} \partial f_d(\boldsymbol{x})/\partial x_d = \text{Tr}(\boldsymbol{J})$.

The vector field $u_t(\phi_t(\boldsymbol{x}))$ is often modeled without $\phi_t(\boldsymbol{x})$ invertability requirement by an arbitrary neural network $v_{\boldsymbol{\theta}}(t, \boldsymbol{x}_t)$ with the learnable weight vector $\boldsymbol{\theta}$. Then, the continuity equation in (2) for (1) neural ODE can be written using the instantaneous change of variables [Chen et al., 2018] as

$$d \log p_t(\boldsymbol{x}_t)/dt + \text{Tr}(\partial v_{\boldsymbol{\theta}}(t, \boldsymbol{x}_t)/\partial \boldsymbol{x}_t^T) = 0. \tag{3}$$

The (3) neural ODE can be solved both for a point $\boldsymbol{x}_0$ and the log-likelihood change as integration

$$\begin{bmatrix} \boldsymbol{x}_0 \\ \log(p_1/p_0) \end{bmatrix} = \int_{t=1}^{t=0} \begin{bmatrix} v_{\boldsymbol{\theta}}(t, \boldsymbol{x}_t) \\ -\text{Tr}(\partial v_{\boldsymbol{\theta}}(t, \boldsymbol{x}_t)/\partial \boldsymbol{x}_t^T) \end{bmatrix} dt, \text{ and initially } \begin{bmatrix} \boldsymbol{x}_t \\ \log(p_1/p_t) \end{bmatrix} = \begin{bmatrix} \boldsymbol{x}_1 \\ 0 \end{bmatrix}. \tag{4}$$

Then, the CNF maximizes likelihood (MLE) during training and uses its estimate at the evaluation as

$$\arg\max_{\boldsymbol{\theta}} \mathcal{L}_{\text{MLE}} := \log \hat{p}_1 = \log p_0 - \int_{t=1}^{t=0} \text{Tr}\left(\partial v_{\boldsymbol{\theta}}(t, \boldsymbol{x}_t)/\partial \boldsymbol{x}_t^T\right) dt, \tag{5}$$

where $\log p_0$ is the likelihood of a point $\boldsymbol{x}_0$ from (4) when evaluated using a known prior $q(\boldsymbol{x}_0)$.

**Flow matching training.** Typically, solving the ODE (4) for the MLE objective (5) is computationally expensive [Grathwohl et al., 2019, Zhuang et al., 2020]. The FM framework [Lipman et al., 2023] proposes an alternative objective that regresses the $v_{\boldsymbol{\theta}}(t, \boldsymbol{x}_t)$ to $u_t$ by conditioning the latter by a vector $\boldsymbol{z} = \boldsymbol{x}_1$. This has been extended by the conditional FM (CFM) framework [Tong et al., 2024] where the $u_t(\boldsymbol{x}|\boldsymbol{z})$ and $p_t(\boldsymbol{x}|\boldsymbol{z})$ are conditioned on a more general $\boldsymbol{z} \sim q(\boldsymbol{z})$ such that the marginal probability density path and the corresponding marginal vector field are defined as

$$p_t(\boldsymbol{x}) = \int p_t(\boldsymbol{x}|\boldsymbol{z})q(\boldsymbol{z})\, d\boldsymbol{z}, \text{ and } u_t(\boldsymbol{x}) = \mathbb{E}_{\boldsymbol{z} \sim q(\boldsymbol{z})}\left[u_t(\boldsymbol{x}|\boldsymbol{z})p_t(\boldsymbol{x}|\boldsymbol{z})/p_t(\boldsymbol{x})\right]. \tag{6}$$

The Gaussian conditional probability path in (6) has the unique conditional vector field such that

$$p_t(\boldsymbol{x}|\boldsymbol{z}) = \mathcal{N}(\boldsymbol{x} \,|\, \mu_t(\boldsymbol{z}), \sigma_t(\boldsymbol{z})^2 \boldsymbol{I}) \implies u_t(\boldsymbol{x}|\boldsymbol{z}) = (\boldsymbol{x} - \mu_t(\boldsymbol{z}))\,\sigma_t'(\boldsymbol{z})/\sigma_t(\boldsymbol{z}) + \mu_t'(\boldsymbol{z}), \qquad (7)$$

where the mean $\mu_t(\boldsymbol{z})$ and the standard deviation $\sigma_t(\boldsymbol{z})$ functions parameterize the path $p_t(\boldsymbol{x}|\boldsymbol{z})$.

Finally, the CFM objective for the simplified CNF training using (7) result can be written as

$$\arg\min_{\boldsymbol{\theta}} \mathcal{L}_{\mathrm{CFM}} := \mathbb{E}_{t\sim\mathcal{U}(0,1),\boldsymbol{x}_t\sim p_t(\boldsymbol{x}|\boldsymbol{z}),\boldsymbol{x}\sim\mathcal{N}(\mathbf{0},\boldsymbol{I}),\boldsymbol{z}\sim q(\boldsymbol{z})}\|v_{\boldsymbol{\theta}}(t,\boldsymbol{x}_t) - u_t(\boldsymbol{x}|\boldsymbol{z})\|^2. \qquad (8)$$

Recently proposed $\mu_t(\boldsymbol{z})$ and $\sigma_t(\boldsymbol{z})$ for interpolation of Gaussian conditional probability path $p_t(\boldsymbol{x}|\boldsymbol{z})$ in (7) are given in Table 1. The above CFM framework is inherently limited by the (6) interpolation assumption and (7) path choice. To overcome this, Chen and Lipman [2024] extend the Euclidean FM to Riemannian manifolds. Kapusniak et al. [2024] introduce the metric FM that learns parameterized interpolants. There are attempts to support FM for categorical data Stark et al. [2024], Cheng et al. [2024], Campbell et al. [2024] and mixed categorical-continuous data [Dunn and Koes, 2024]. In this paper, we are motivated by a different perspective: *is it feasible to avoid the interpolation in FM framework with a reasonable computational cost?*

## 3   The Proposed Interpolant-free Dual Flow Matching

**Dual CNF via a reverse vector field.** Let's extend the CNF framework that is defined in (1-4). We introduce a *dual CNF* where its first part, implemented as the above non-invertible $v_{\boldsymbol{\theta}}(t,\boldsymbol{x}_t)$, approximates the vector field $u_t(\phi_t(\boldsymbol{x}))$. In addition, we employ an extension $v_{\boldsymbol{\lambda}}(t,\boldsymbol{y}_t)$ with learnable parameters $\boldsymbol{\lambda}$ that models a *reverse vector field* model $u_t(\phi_t^{-1}(\boldsymbol{y}))$. In other words, there is the forward transformation $\boldsymbol{x}_t = \phi_t(\boldsymbol{x})$ and the inverse transformation $\boldsymbol{y}_t = \phi_t(\boldsymbol{y}) = \phi_t^{-1}(\boldsymbol{x})$ of the bijective map $\phi_t$.

Then, we can reformulate the equations (4) for $v_{\boldsymbol{\lambda}}(t,\boldsymbol{y}_t)$ model with minor modifications. The neural ODE in (3) can be solved in reverse simultaneously for a point $\boldsymbol{x}_1$ and the log-likelihood change with the initial condition $\boldsymbol{y}_t \sim q(\boldsymbol{x}_0)$ and $\log(p_0/p_t) = 0$ as integration

$$\begin{bmatrix} \boldsymbol{x}_1 \\ \log(p_0/p_1) \end{bmatrix} = \int_{t=0}^{t=1} \begin{bmatrix} v_{\boldsymbol{\lambda}}(t,\boldsymbol{y}_t) \\ -\mathrm{Tr}\left(\frac{\partial v_{\boldsymbol{\lambda}}(t,\boldsymbol{y}_t)}{\partial \boldsymbol{y}_t^T}\right) \end{bmatrix} dt. \qquad (9)$$

Figure 1: The CFM (top) regresses the Gaussian-interpolated forward vector field by a neural network with the *affine transformation* $\phi_t(\boldsymbol{x})$. Our DFM (bottom) has two neural networks with the *free-form transformations* with only the bijectivity objective $\boldsymbol{x}_t = \phi_{\boldsymbol{\lambda}}^{-1}(\phi_{\boldsymbol{\theta}}(\boldsymbol{x}_t))$ for an arbitrary vector field and a probability path.

Interestingly, the (9) approach with the modified maximum likelihood $\mathcal{L}_{\mathrm{MLE}}(\boldsymbol{\lambda})$ is known in the DNF literature as a reverse divergence objective [Papamakarios et al., 2021]. When the target data $p_1$ cannot be analytically evaluated, the (9) is impractical for CNF training with the MLE objective.

**Interpolant-free DFM.** On the other hand, the proposed dual CNF with the reverse model in (9) can be used for the interpolant-free flow matching. Instead of the less expressive *affine transformation* ($\phi_t(\boldsymbol{x}|\boldsymbol{z}) = \mu_t(\boldsymbol{z})+\sigma_t\boldsymbol{x}, \boldsymbol{x}\sim\mathcal{N}(\mathbf{0},\boldsymbol{I})$) induced by the Gaussian interpolation (7), the proposed DFM only requires *bijectivity of the free-form transformations* $\phi_t$ and $\phi_t^{-1}$ produced by, correspondingly, the forward $v_{\boldsymbol{\theta}}(t,\boldsymbol{x}_t)$ and the reverse $v_{\boldsymbol{\lambda}}(t,\boldsymbol{y}_t)$ vector field models.

Then, the proposed dual CNF with the bijective $\phi_t$ can be expressed as ODEs expressed by

$$\begin{cases} d\phi_t(\boldsymbol{x})/dt = u_t(\phi_t(\boldsymbol{x})) = v_{\boldsymbol{\theta}}(t,\boldsymbol{x}_t) \\ d\phi_t(\boldsymbol{y})/dt = u_t(\phi_t(\boldsymbol{y})) = v_{\boldsymbol{\lambda}}(t,\boldsymbol{y}_t). \end{cases} \qquad (10)$$

Assuming the $\phi_t(\boldsymbol{y}) = \phi_t^{-1}(\boldsymbol{x})$ bijectivity in a neighborhood of $t$ for $\boldsymbol{x}$ and $\boldsymbol{y}$, (10) can be rewritten using the univariate inverse function theorem by substituting the top to bottom as

$$d\phi_t^{-1}(\boldsymbol{x})/dt = 1/\left(d\phi_t(\boldsymbol{x})/dt\right) \implies \mathrm{diag}\left(v_{\boldsymbol{\theta}}(t,\boldsymbol{x}_t) \odot v_{\boldsymbol{\lambda}}(t,\boldsymbol{y}_t)\right) = \boldsymbol{I}. \qquad (11)$$

The (11) objective can be achieved by minimizing the cosine distance between two unit vectors as

$$\arg\min_{\boldsymbol{\theta},\boldsymbol{\lambda}} \mathcal{L}_{\text{DFM}} := \mathbb{E}_{t\sim\mathcal{U}(0,1),\boldsymbol{x}_t\sim\hat{q}(\boldsymbol{x}_1),\boldsymbol{y}_t\sim q(\boldsymbol{x}_0)} \text{dist}_{\cos}\left(v_{\boldsymbol{\theta}}(t,\boldsymbol{x}_t), v_{\boldsymbol{\lambda}}(t,\boldsymbol{y}_t)\right), \qquad (12)$$

where this loss with vector normalization is more numerically stable in practice.

Once the (12) loss is minimized, the density estimation i.e. $\log\hat{p}_1$ can be performed using the conventional MLE approach (5) without the extension $v_{\boldsymbol{\lambda}}(t,\boldsymbol{y}_t)$. On the other hand, the extension can be used to improve $\log\hat{p}_1$ by integrating (9) from $t=1$ to $t=0$. We use the latter strategy. While we yet to accomplish sampling experiments, we expect DFM to outperform previous FM methods due to the enforced bijectivity which is a common issue in ODEs [Gholami et al., 2019].

## 4 Experiments

**Benchmark.** We employ real-world SMAP [Hundman et al., 2018] time series benchmark for unsupervised anomaly detection. The soil moisture active passive satellite (SMAP) dataset contains soil samples and telemetry information from the Mars rover with 135K and 428K data points in the training (without anomalies) and test sets, respectively. SMAP data has 25 data dimensions collected from 55 entities. We follow Su et al. [2019] and transform the regression task into a classification task using sliding windows (window size = 8) and replication padding [Tuli et al., 2022].

**Flow models.** We report experimental results for the Glow-type DNF [Kingma and Dhariwal, 2018] from [Gudovskiy et al., 2024] with the state-of-the-art baselines. Second, we experiment with the vanilla CNF from Section 2 and the CNF trained using the CFM framework i.e., the FM from [Lipman et al., 2023] and I-CFM from [Tong et al., 2024]. All CNF models have exactly the same U-Net architecture [Ronneberger et al., 2015], learnable $\mathcal{N}(\boldsymbol{\mu}, \boldsymbol{\sigma}^2\boldsymbol{I})$ prior and identical evaluation using (5).

**Evaluation.** We follow Su et al. [2019] and report precision (P), recall (R), AuC and $F_1$ score. We provide results when we solve ODE using the fixed-step (F) Euler method with 4 steps and the variable-step (V) Dopri5 method (atol=1e-1, rtol=1e-2) from the [Zhuang et al., 2021] library. We use the Hutchinson stochastic estimator of the Jacobian matrix trace [Hutchinson, 1990].

**Quantitative results.** We compare flow models to other popular baselines: OmniAnomaly [Su et al., 2019], CAE-M [Zhang et al., 2021], TranAD [Tuli et al., 2022]. It is common in these baselines to train and evaluate a separate model for each SMAP entity. In contrast, all our flow models use a single model for all entities in Table 2 i.e. they are entity-unconditional.

We can derive several important conclusions from Table 2 results. First, continuous-time normalizing flows models, if properly trained and evaluated, are able to outperform discrete-time normalizing flows as well as other non-flow models in this density estimation task. Second, recent integration-free FM training methods using (8) perform similarly or better than the CNF trained by computationally-expensive

Table 2: SMAP unsupervised anomaly detection. The **best** and the second best metrics, %.

| Model | $\int$ | P | R | AUC | $F_1$ |
|---|---|---|---|---|---|
| OmniAnom. | ✗ | 81.3 | 94.2 | 98.9 | 87.3 |
| CAE-M | ✗ | 81.9 | 95.7 | 99.0 | 88.3 |
| TranAD | ✗ | 80.4 | 99.9 | 99.2 | 89.2 |
| Glow DNF | ✗ | 87.4 | 84.9 | 91.6 | 86.1 |
| Base CNF | F | 87.5 | 98.8 | 98.4 | 92.8 |
| FM | F | 88.2 | 98.9 | 98.5 | 93.3 |
| I-CFM | F | 88.0 | **99.2** | 98.6 | 93.3 |
| DFM (ours) | F | **94.7** | 98.1 | **98.7** | **96.4** |
| Base CNF | V | 86.5 | 91.9 | 94.9 | 89.1 |
| FM | V | 87.4 | **99.6** | **98.7** | 93.1 |
| I-CFM | V | 89.3 | 98.2 | 98.3 | 93.6 |
| DFM (ours) | V | **89.7** | 98.9 | 98.6 | **94.1** |

MLE from (5). Third, the proposed DFM significantly outperforms prior FM methods with only the $2\times$ complexity increase. In particular, DFM increases the non-saturated metrics such as precision and $F_1$ score by, correspondingly, 6.5 (88.2% →94.7% ) and 3.1 (93.3% →96.4% ) percentage points.

## 5 Conclusions

In this paper, we analyzed limitations of the interpolation-based flow matching framework that allows to efficiently train a CNF model. To address the limitations, we proposed the interpolant-free dual flow matching method. Our experiments with the SMAP benchmark showed that our DFM achieves state-of-the-art results for the entity-unconditional unsupervised anomaly detection. In future, the DFM objective (11) and the practical loss (12) can further be extended to multivariate case.

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
