# OpenReview forum: "DFM: Interpolant-free Dual Flow Matching"
_NeurIPS.cc/2024/Workshop/UniReps — UniReps_

### Official Review · Reviewer_A7Wh · 2024-09-27
**Dual Flow Matching to improve continuous normalizing flows by challenging the Gaussian probability path assumption in Flow Matching approach**

**Rating:** 7
**Confidence:** 2

**Review:**

# Strengths
1. **Strong Theoritical Foundation**: The mathematical introduction for CNF and FM is very clear. It is easy to why interpolation is the bottleneck and how DFM fills this gap.
2. **Proposed Approach**: The novel objective function of enforcing bijectivity is mathematically grounded and straightforward.
3. **Experiments**: The empirical experiments demonstrate the consistent improved performance of DFM over the baselines.
4. **Overall Paper**: The paper is well written, with clear notations and definitions. It was easy to follow.

# Weaknesses
I think the paper is solid for the extended abstract track.

# Suggestions
1. It would be nice to have some visualizations of the flow, maybe in Appendix.

---

### Official Review · Reviewer_GfKU · 2024-09-28
**An interesting extension of the flow matching framework**

**Rating:** 7
**Confidence:** 3

**Review:**

DFM: Interpolant-free Dual Flow Matching

The paper proposes an objective to train flow matching models without needing to pre-specify a probability path, by parameterizing both the forward path and the reverse path.

Strength: 1. The paper provides an interesting extension of the flow matching framework. 2. Some empirical evidence was provided.

Weakness: 1. Parameterizing both the forward path and the reverse path leads to some computational overhead. 2. It is not clear how the method performs in some more complex cases, e.g. in high dimensional generative modeling tasks. 3. One benefit of flow matching is, by specifying the desired probability path, it is possible to impose some desired properties on the model, for instance preferring straight paths; see e.g. [1]. It is not clear what the learned probability paths of the proposed method are like.

[1] Improving and generalizing flow-based generative models with minibatch optimal transport, Tong et al.

---

### Official Review · Reviewer_y88F · 2024-10-04
**Good idea to use a reverse network.**

**Rating:** 6
**Confidence:** 3

**Review:**

This paper proposes using a neural network to replace the fixed reverse process during Flow Matching. Experiments on anomaly detection show that it is comparable to baseline Conditional Normalizing Flow and Flow Matching methods. The precision and the F1 score are significantly higher on the SMAP dataset. I still have several questions:
1. Regarding motivation, I don't understand the logic of "solving an ODE is expensive". Isn't it more expensive to learn an additional neural network? If so, perhaps the last sentence in Section 2 is more suitable.
2. I don't understand the advantage of the cosine distance. Why does equation 12 use cosine distance and CFM in equation 8 use L2 distance? A more detailed discussion of selecting a good metric could be extended.
3. The reproducibility and fairness of comparisons are potential issues. More experimental settings should be provided. For example, can DFM be compared with baselines using the L2 metric?
Overall, the paper is well presented. More datasets can be studied to ensure the method's generality and increase its significance.

---

### Official Review · Reviewer_Cfn1 · 2024-10-06
**This paper proposes an interpolant-free flow matching method, DFM, using a reverse vector field and conducts experiments to evaluate its effectiveness.**

**Rating:** 8
**Confidence:** 4

**Review:**

##### Summary of Strengths

- The writing is clear and easy to follow. The related works are well-organized.
- The proposed DFM method is grounded in solid theoretical proof.

##### Summary of Weaknesses

- There are no significant weaknesses. It would be beneficial to compare the proposed method with other flow matching techniques in the domain of image generation.

---

### Decision · Program_Chairs · 2024-10-10

**Decision:**

Accept

**Comment:**

In light of the positive reviewers' feedback and relevancy of the submission, we are pleased to accept this paper for presentation at UniReps 2024. We kindly ask the authors to incorporate the reviewers' suggestions and feedback in the final camera-ready version of the manuscript.